# SegGraph: Leveraging Graphs of SAM Segments for Few-Shot 3D Part Segmentation

**Yueyang Hu[1], Haiyong Jiang[1]\*, Haoxuan Song[1], Jun Xiao[1]\*, Hao Pan[2]**
[1]School of Artificial Intelligence, University of Chinese Academy of Sciences
[2]School of Software, Tsinghua University

{huyueyang23, songhaoxuan24}@mails.ucas.ac.cn
{haiyong.jiang, xiaojun}@ucas.ac.cn
haopan@tsinghua.edu.cn

## Abstract

This work presents a novel framework for few-shot 3D part segmentation. Recent advances have demonstrated the significant potential of 2D foundation models for low-shot 3D part segmentation. However, it is still an open problem that how to effectively aggregate 2D knowledge from foundation models to 3D. Existing methods either ignore geometric structures for 3D feature learning or neglects the high-quality grouping clues from SAM, leading to under-segmentation and inconsistent part labels. We devise a novel SAM segment graph-based propagation method, named SegGraph, to explicitly learn geometric features encoded within SAM's segmentation masks. Our method encodes geometric features by modeling mutual overlap and adjacency between segments while preserving intra-segment semantic consistency. We construct a segment graph, conceptually similar to an atlas, where nodes represent segments and edges capture their spatial relationships (overlap/adjacency). Each node adaptively modulates 2D foundation model features, which are then propagated via a graph neural network to learn global geometric structures. To enforce intra-segment semantic consistency, we map segment features to 3D points with a novel view-direction-weighted fusion attenuating contributions from low-quality segments. Extensive experiments on PartNet-E demonstrate that our method outperforms all competing baselines by at least 6.9% mIoU. Further analysis reveals that SegGraph achieves particularly strong performance on small components and part boundaries, demonstrating its superior geometric understanding. The code is available at: https://github.com/YueyangHu-2000/SegGraph.

## 1   Introduction

3D part segmentation is a fundamental task in computer vision and graphics with broad implications for shape analysis [1], 3D modeling [2, 3], and robotic manipulations [4, 5]. For instance, a user can edit a 3D shape based on part components and embodied agents need to interact with diverse object parts for different manipulations, such as identifying a drawer's handle or a bottle's neck. Many of these real-world applications involve novel shapes, and it is essential to achieve high-quality 3D part segmentation with only a few number of annotations.

2D foundation models (FMs) with powerful generalization capacities have brought new possibilities to part segmentation. Nevertheless, the modality gap between 2D images and 3D geometric shapes poses a fundamental challenge for their direct deployment in 3D domains. A prevalent strategy [6, 7]

---

\*Joint Corresponding Authors. Haiyong is the Project Lead.

39th Conference on Neural Information Processing Systems (NeurIPS 2025).

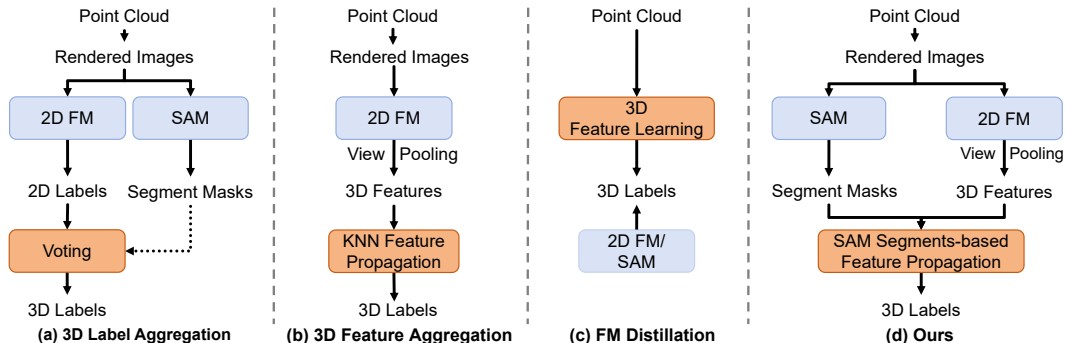

Figure 1: Architecture comparisons of different methods. (a) 3D label aggregation-based methods [6, 18, 11, 12, 7], (b) 3D feature aggregation-based methods [16, 17], (c) distillation-based methods [13, 14, 15] to distill 2D knowledge from foundation models, and (d) ours with an emphasis on SAM segments and SAM-segments-based feature propagation.

is to render a 3D shape into multi-view images, then leverage 2D foundation models such as GLIP [8] or Diffusion [9] for image segmentation, and finally aggregate 2D labels to the 3D domain with voting (see Fig. 1a). During the process, segmentation foundation models, e.g., SAM [10], can further enhance the semantic awareness for finer part segmentation [11, 12, 7]. However, simply aggregating 2D labels to 3D domains overlooks the geometric structures of 3D inputs, resulting in under-segmentation of shape parts and inconsistency among neighbor points.

3D distillation from foundation models [13, 14, 15] and 3D aggregation-based feature learning [16, 17] take a step forward and learn 3D features with geometry awareness (see Fig. 1b,c). Distillation-based methods typically learn geometric features by transferring knowledge from foundation models but rely on large-scale 3D shapes for good performance. 3D feature aggregation-based methods aggregate multi-view image features from foundation models and propagate fused features via KNN-based point structures. However, simple KNN propagation struggles to encode complex geometric patterns and does not take into account grouping clues from SAM. Moreover, these two schemes involve downsampling 3D points (about 100 times for PartNet-E) or using voxel-based 3D feature encoding, leads to ambiguous features for boundary points and small components.

We propose a novel framework, called **SegGraph**, for part segmentation of 3D point clouds. We develop the method based on three key insights. First, SAM's high-quality segmentation provides consistent intra-segment grouping cues for 3D points, effectively reducing misclassification errors near boundaries. These segments can efficiently encode 3D geometric structures with less than 1000 SAM segments for each shape, avoiding input downsampling. Second, spatial relationships among segments offers two critical priors for part segmentation: (1) overlapping segments across different views typically share the same part label, (2) adjacent segments naturally preserve part boundary information. Third, the quality of SAM segments correlates with the viewing direction of an image, exhibiting under-segmentation of small parts in challenging views.

To implement these three observations, we introduce a segment graph propagation-based part learning framework, as illustrated in Fig. 2. First, we encode 3D point features through multi-view feature pooling from 2D foundation model outputs. Simultaneously, we generate view-consistent 3D segments by aggregating SAM-based 2D segmentations across all views. We then model both the spatial relationships (overlapping/adjacent) between segments and the constituent relationships between points and segments. The segment graph takes 3D segments as nodes and represent spatial relationships as nodes. Segment features encoded from its constituent points' aggregated features are propagated via a graph neural network for geometric structure-aware feature encoding. Thereafter, we enhance 3D point features with segment features with a view quality-aware feature unpooling, where points with a normal off a view direction for obtaining the SAM segment are assigned with lower importance.

Extensive experiments demonstrate that SegGraph can significantly outperform competing methods by a large margin of 6.9% mIoU on the PartNet-E dataset [6]. SegGraph is quite extensible and can leverage different kinds of foundation models with an improvement of at least 4% on the PartNet-E dataset.

## 2   Related Work

**Supervised 3D Segmentation.** Supervised 3D segmentation has been widely studied in recent years, facilitated by the availability of annotated datasets [19, 20, 21, 22, 23, 24]. Prior works have proposed a variety of architectures tailored to 3D data, including point-based models (e.g., PointNet [25] and its variants [26, 25, 27, 28] ), volumetric CNNs [29, 30, 31], graph convolutional networks [32, 33, 34] and Transformer-based methods [35, 36, 37]. These methods are typically trained with large amounts of manually annotated 3D data. A particularly challenging sub-task is 3D part segmentation that predicts semantic part labels for a shape. PartNet [23] and DeepGCNs [38] achieves part segmentation via a graph convolutional network (GCN)-based model to learn local geometric features. CSN [39] improves part segmentation by introducing a cross-shape attention mechanism that captures interactions among different shapes in 3D point clouds. However, most of these methods rely on dense part annotations for good performance. Some works explore multi-prototype networks with attention [40] and learning part-specific probability spaces via template morphing and density estimation [41] for few-shot part segmentation.

**3D Part Segmentation with Foundation Models.** Recent advancements of foundation models [42, 10, 43] have shown remarkable capabilities across diverse tasks. A simple yet effective approach to exploit 2D foundation models for 3D tasks is to render multi-view projections of the 3D data, feed the rendered images into a 2D foundation model, and subsequently aggregate the model outputs back to 3D via voting. Pioneering efforts like PartSLIP [6] and SATR [18] follow this scheme and utilize GLIP [8] for open-vocabulary segmentation. PartSLIP++ [11] and PartSTAD [12] further utilize detections of GLIP as the visual prompts for SAM [10] for better part masks, while 3-by-2 [7] constructs multiview labels based on DIFT [9] and aggregates 2D labels to 3D based on multiview masks from SAM [10]. However, aggregating 2D segmentation into 3D entirely overlooks the inherent geometric structure of the 3D data and consistent point segmentations among neighbors, leading to under-segmentation and noisy parts. COPS [44] and [17] address this problem by directly aggregating 2D features from foundation models to 3D and propagating 3D features via superpoint-based pooling and KNN-based feature smoothing for superpoints. Despite better results, feature aggregation with simple KNN smoothing disables complex 3D context learning and geometric superpoints are often inconsistent with part grouping. Another line of work [13, 14, 15] leverages a distillation-based framework. For example, PartDistill [13] presents a bidirectional distillation between predicted 2D labels and 3D labels from a native 3D network. SAMPart3D [14] and PartField [15] construct training pairs for contrastive learning based on SAM segmentation. These distillation methods require a large amount of 3D data to learn generalizable 3D features. This work combines the benefits of both SAM segmentations and light-weight 3D feature aggregation with shape structural context encoding for easy adaption to few-shot novel shapes.

## 3   Methodology

Our goal is to predict a part label for each point of a given point cloud with only low-shot training samples for novel shapes. We approach this problem with a graph-based feature aggregation network as illustrated in Fig. 2. The method encodes point-wise features using an off-the-shelf foundation model (Sec. 3.1) and generates a list of segment masks based on SAM segmentation (Sec. 3.2). Then we construct a SAM-segment-based graph to encode different kinds of relations among different segments and 3D points (Sec. 3.3).

### 3.1   Feature Encoding for 3D Points

To harness the power of image foundation models, such as DINOv2 [43] and CLIP [42], our approach renders a point cloud input into multi-view images from $M$ predefined camera views. We denote rendered images as $\{I_m \in \mathbb{R}^{H \times W \times 3}\}_{m=1}^M$. The rendering process follows PartSLIP [6], incorporating occlusion culling [45] to eliminate obscured points. Next, we feed each view image to an image foundation model (DINOv2 in our implementation) to extract image features. However, the resolution of output image feature maps ($\frac{W}{14} * \frac{H}{14}$) is usually lower than the input image. Therefore, image feature maps are upsampled to the original image resolution using bicubic interpolation and are mapped from 768 channels to 96 channels with a linear layer. We can obtain the 3D feature $F^p$ for each point by taking the average of its projected image features at views where the point is not occluded.

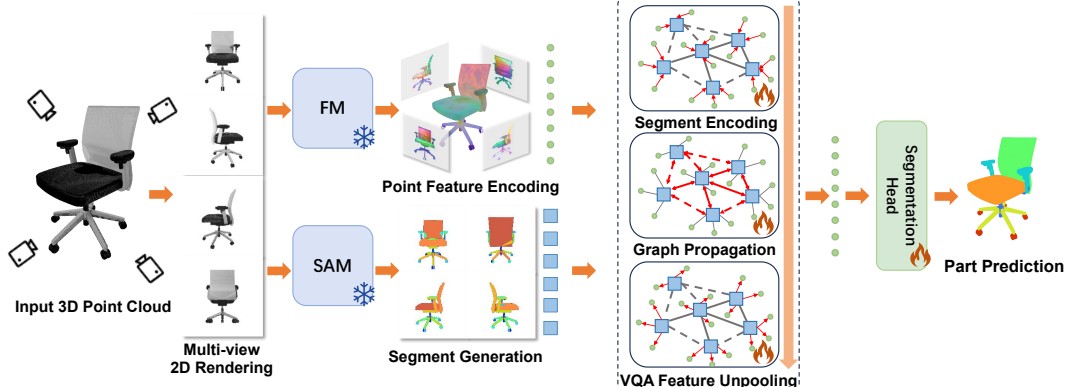

Figure 2: The overview of the pipeline. Given a 3D point cloud, we render it into multi-view images. We extract individual point features by pooling foundation model features (DINOv2) of rendering views. At the same time, multi-view images are fed to SAM for segment generation. Point features and segments are sent to a segment graph to learn geometric features and segment relations for part prediction.

## 3.2 Segment Generation

In light of the high-quality segmentation masks of SAM [10], we further leverage these masks to enhance the quality of part segmentation. While the semantic granularity of SAM's predictions may vary across views, these masks nevertheless provide reliable grouping clues (see the colored segments in Fig. 3). Our method processes each rendered image using SAM to produce an initial set of segmentation masks. Following the approach in [7], we subsequently decompose overlapping masks under a single view into non-overlapping, over-segmented regions. This decomposition effectively resolves semantic confusions for 3D points whose projections fall within multiple overlapping masks of varying granularity and semantics. Afterwards, all 3D points that project within the same 2D segmentation mask form a corresponding segment. The set of segments is denoted as $\mathcal{S}$.

## 3.3 Segment-based Feature Refinement via Graph Propagation

Given 3D point features $\boldsymbol{F}^p$ and segments $\mathcal{S}$, we further exploit the grouping cues from individual segments, along with their underlying geometric structures, to enhance 3D features. The key motivations are as follows: 1) points within the same segment typically share the same part label; 2) overlapping segments constructed from different views likely correspond to the same part label; 3) adjacent segments from the same or different views encode the holistic 3D structure and delineate possible part boundaries. To compile these ideas into a framework, we first encode segment features, then propagate them through overlapping segments and adjacent segments, and finally map segment features to individual point for part segmentation.

**Segment Encoding.** A straightforward way to compute segment features is to average the features of their member points. However, simple pooling fails to account for the varying semantic importance of individual points based on their geometric attributes, including their spatial distribution relative to the segment centroid and local surface normal. Inspired by [46], we instead learn adaptive point contributions with a geometric feature encoding module. For each point $\boldsymbol{p}_j$ within a segment, the module calculates its normalized relative position $\boldsymbol{p}_j^r$ w.r.t. the centroid $\boldsymbol{c}_i$ of a segment $\mathcal{S}_i$ as follows:

$$\boldsymbol{p}_j^r = \frac{\boldsymbol{p}_j - \boldsymbol{c}_i}{\max\limits_{k \in \mathcal{S}_i}(\boldsymbol{p}_k) - \min\limits_{k \in \mathcal{S}_i}(\boldsymbol{p}_k)}. \tag{1}$$

Then we compute the local geometric feature $\boldsymbol{F}_j^l \in \mathbb{R}^C$ with a multi-layer perceptron that takes as input the concatenation of point's normal and its normalized position $\boldsymbol{p}_j^r$. Afterwards, we obtain the segment feature $\boldsymbol{F}^s \in \mathbb{R}^C$ by applying max pooling to the local geometric features $\boldsymbol{F}_j^l$ of all points within the segment. The segment feature $\boldsymbol{F}^s$ is further refined through an attentively weighted aggregation of point features $\boldsymbol{F}_j^p$, where attentive weights are computed with an additive attention [47] on $\boldsymbol{F}_j^l$ and $\boldsymbol{F}^s$.

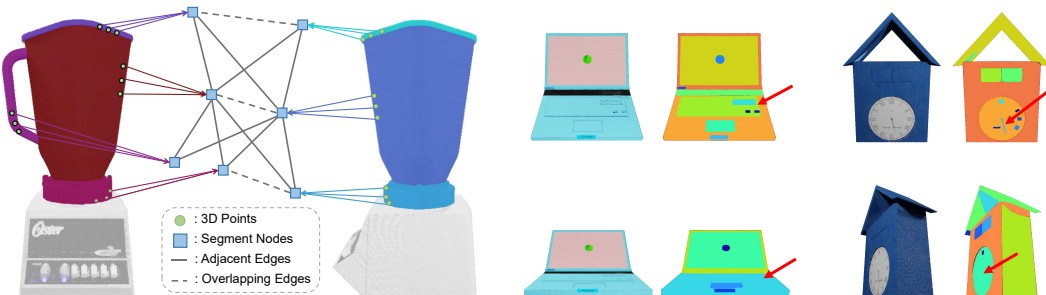

Figure 3: An illustration of graph construction. Features of 3D points (green dots) are first aggregated to segment nodes (blue boxes).

Figure 4: Two examples for the impacts of the rendering views on the quality of SAM segments. For each group, left: input, right: SAM segments.

**Building a SAM Segment-based Graph.** Denote the graph as $\mathcal{G} = (\mathcal{V}, \mathcal{E})$ with nodes $\mathcal{V}$ representing 3D segments and edges $\mathcal{E}$ encoding segment relationships. The segment relationships include 3D adjacent relations $\mathcal{E}_a$ and overlapping relations $\mathcal{E}_o$. An overlapping relationship exists between two segment nodes in different views if their corresponding 3D points have significant overlaps (mIoU larger than $10\%$). An adjacent relation between two segments is constructed if their covered points have rare overlaps but are close enough (with a minimal distance between points of two segments less than 0.01 units in the normalized space). Note that overlapping relations and adjacent relations are mutually exclusive. We show an example of a segment graph in Fig. 3. In this example, overlapping segments (connected by dashed lines) share identical part labels, while adjacent segments (connected by solid lines) demarcate part boundaries. This graph structure is analogous to an atlas in differentiable manifold mapping the 3D shape consistently. Specifically, each segment corresponds to a chart, and transitions between charts are edges derived from the adjacency and overlap relationships among segments; the collection of all charts and their connecting edges constitutes an atlas. In this way, our segment graph provides a structure for analyzing the semantics of object shapes with geometric consistency.

**Feature Propagation via the Segment Graph.** After building the segment graph, we propagate segment features through the edges using GATv2 [48]. We feed the segment feature $\boldsymbol{F}^s$ to a three GATv2 network layer. Given the distinct nature of overlapping and adjacent relationships, we implement two GATv2 networks with separate weight parameters to model these relationships, respectively. Then we concatenate the output features of two GATv2 at each layer and feed them to a MLP as the node features for the next layer.

**Viewing Quality-Aware Feature Unpooling.** Subsequently, we propagate segment features to member 3D points to enforce label consistency within segments. This propagation requires careful feature fusion since points usually belong to multiple segments with varying reliability. A naive averaging approach is suboptimal as segments from some challenging views are usually of low quality. As shown in Fig. 4, under certain challenging viewpoints, parts such as laptop keyboards and clock components are not well segmented by SAM. In contrast, under more favorable viewpoints, SAM is able to segment these parts accurately. As the quality of segment masks often depends on the viewing angle, we devise a viewing quality-aware unpooling module for this purpose. This module estimates the quality based on the point normal $\boldsymbol{n}_j$ and the camera view direction:

$$\boldsymbol{w}_{ij}^v = |\boldsymbol{n}_j \cdot \frac{\boldsymbol{p}_j - \boldsymbol{c}_i}{\|\boldsymbol{p}_j - \boldsymbol{c}_i\|}|, \tag{2}$$

where $\boldsymbol{c}_i$ is the camera position for extracting segment $i$ and $\boldsymbol{n}_j$ is the normal direction of point $j$. The quality score $\boldsymbol{w}_{ij}^v$ is further refined with a learnable MLP with a softmax to reweighing the quality scores. The final 3D point features combine the original point features and a weighted fusion of segment features, and then the features are fed to the segmentation head with a two-layer MLP to produce the logits $\hat{\boldsymbol{Y}}$:

$$\boldsymbol{F}_i^{p\prime} = \boldsymbol{F}_i^p + \sum_{i \in \mathcal{S}_j} \boldsymbol{w}_{ij}^v \cdot \boldsymbol{F}_j^S, \qquad \hat{\boldsymbol{Y}}_i = \text{MLP}(\boldsymbol{F}_i^{p\prime}). \tag{3}$$

For the network training, we use cross-entropy as the objective to train the network with few-shot examples.

## 4    Experimental Results

We evaluate the effectiveness of our method on two benchmark datasets: PartNet-Ensemble (PartNet-E) [6] and ShapeNetPart [49]. PartNet-E, proposed by PartSLIP [6], consists of 1,906 shapes across 45 categories with RGB colors. The dataset splits follow that of [12]. ShapeNetPart contains 31,963 shapes spanning 16 categories but lacks color information. We adopt the mIoU as the evaluation metric. We evaluate our method on the official testing set using a few-shot setting, where 8-shot examples are sampled from the training set for all experiments. We trained our method three times and reported the average mIoU and the standard deviations to reflect the influence of training variances using different seeds.

### 4.1    Comparisons on Part Segmentation

**Baselines.**    On the PartNet-E dataset, we compare SegGraph with fully supervised approaches [37, 26, 50] as well as state-of-the-art few-shot methods based on 2D foundation models including label aggregation-based methods [6, 11, 12, 7], and distillation-based methods [13, 15]. The fully supervised methods were trained on 28K objects from 17 overlapping categories in PartNet [23] that overlap with PartNet-E, along with the few-shot training shapes. The few-shot baselines were trained solely on the few-shot set. For the zero-shot method PartField, which successfully distills part-level 3D representations from SAM, we fine-tune an MLP on its extracted features to adapt the model to the few-shot setting. Among the above-mentioned methods, [6, 11, 12, 13] provide testing codes and weight checkpoints for comparisons, while [7] does not have available sources. For methods without sources and that we fail to reproduce, we use their reported results and mark them with *. However, as few-shot methods, e.g., [6, 12, 13], require prompt tuning and do not have training code, we instead compare with PointCLIP [51] and PointCLIPv2 [52] with support of few-shot learning on the ShapeNetPart. Specifically, we replace text embedding for classification in PointCLIP and PointCLIPv2 with a trainable MLP classifier for few-shot example training.

**Results on PartNet-E.** Tab. 1 presents the comparison results on PartNet-E (see the supplementary for per-category results). All foundation model-based methods achieve superior performance to supervised ones (the top three rows), even with only a few-shot training. This suggests the strong capability of foundation models on downstream tasks. We also observe that label aggregation-based methods [6, 11, 12, 7], performs worse than feature aggregation-based methods [7] and distillation-based methods [13, 15], because label aggregation-based methods rely solely on 2D images to obtain labels, completely ignoring the geometric information crucial for 3D part segmentation. Among all methods, our approach outperforms the current state-of-the-art few-shot part segmentation method, PartDistill, by a significant margin with more than $6\%$ gains in mIoU across all 45 categories, achieving the highest performance in 32 of them. Notably, we observe over $10\%$ absolute improvements on categories such as Door and Lamp, which, based on our observations, can be attributed to our method's superior ability to segment relatively small parts such as door handles and light bulbs.

Despite the overall improvement being substantial, different training runs of our method produce varying performance because of random initial seeds. When training the model three times with the same setting, the average standard deviation across the 45 categories is about $1.09\%$.

Since most of these foundation model-based methods use GLIP, to ensure a fairer comparison, in addition to employing DINOv2, we also transformed GLIP into a feature extractor for comparison, as detailed in Sec. 4.2. The results show that when using GLIP as the feature extractor, our method still achieves performance that is only slightly lower than that obtained with DINOv2, as shown in Tab. 1.

Further inspection of segmentation results with large variances reveals that semantically ambiguous 3D segments are prone to errors in the few-shot setting, leading to the variance in different training runs with random initial seeds.

We further inspect the improvements on different sized parts. Benefiting from the over-segmented segments and the segment graph, our method demonstrates significantly improved performance (more than $+20\%$) in segmenting small parts. As illustrated in Tab. 2, we select six representative categories that include small parts such as buttons (in Coffee Machine, Remote), knobs (in Coffee Machine),

Table 1: Few-shot part segmentation results on PartNet-E. The mIoU metric is measured in percentage. The number in the top row bracket counts the number of shape categories. SD is for Stable Diffusion [53]. The methods in the top three rows are trained with both few-shot shapes and PartNet shapes that share overlapping categories (17) with PartNet-E.

| Methods | FM | Overlapping Categories (17) | | | | | | None-Overlapping Categories (28) | | | | | | (45) |
|---|---|---|---|---|---|---|---|---|---|---|---|---|---|---|
| | | Bottle | Chair | Door | Knife | Lamp | Overall | Camera | Dispe. | Kettle | Oven | Suitca. | Overall | Overall |
| PointNet++ [26] | - | 48.8 | 84.7 | 45.7 | 35.4 | 68.0 | 55.3 | 6.5 | 12.1 | 20.9 | 34.4 | 40.7 | 25.0 | 36.5 |
| SoftGroup [50] | - | 41.4 | 88.3 | 53.1 | 31.3 | 82.2 | 50.4 | 23.6 | 18.9 | 57.4 | 13.7 | 18.3 | 31.3 | 38.5 |
| PointNext [37] | - | 68.4 | 91.8 | 43.8 | 58.7 | 64.9 | 59.1 | 33.2 | 26.0 | 45.1 | 37.8 | 13.6 | 45.5 | 50.6 |
| PartSLIP [6] | GLIP | 83.4 | 85.3 | 40.8 | 65.2 | 66.1 | 56.3 | 58.3 | 73.8 | 77.0 | 73.5 | 70.4 | 61.3 | 59.4 |
| PartSLIP++ [11] | GLIP & SAM | 85.5 | 85.3 | 45.1 | 64.3 | 68.0 | 59.7 | 63.2 | 72.0 | 85.6 | 70.3 | 70.0 | 63.5 | 62.1 |
| PartSTAD [12] | GLIP & SAM | 83.6 | 85.3 | 61.4 | 63.8 | 68.4 | 61.4 | 64.4 | 73.7 | 84.2 | 71.9 | 68.3 | 67.1 | 65.0 |
| 3-By-2 [7]* | SD & SAM | 80.9 | 84.4 | 54.4 | 75.1 | 59.5 | 60.4 | 62.6 | 78.2 | 81.5 | 60.0 | 65.2 | 66.5 | 64.2 |
| PartDistill [13]* | GLIP & SAM | 84.6 | 88.4 | 55.5 | 71.4 | 69.2 | 64.6 | 60.1 | 74.7 | 78.6 | 72.8 | 73.4 | 66.7 | 65.9 |
| PartField [15] | SAM | 75.9 | 87.6 | 65.4 | 72.9 | 73.8 | 65.7 | 52.0 | 73.9 | 82.4 | 50.2 | 72.0 | 66.7 | 66.3 |
| SegGraph (ours) GLIP & SAM | | **87.5** (± 1.73) | **88.5** (± 0.69) | **69.5** (± 0.45) | **79.3** (± 1.70) | **84.2** (± 2.19) | **69.1** (± 2.04) | **66.7** (± 0.88) | **74.8** (± 2.39) | **88.2** (± 1.20) | **70.2** (± 3.40) | **72.8** (± 1.54) | **71.5** (± 2.13) | **70.4** (± 2.08) |
| DINOv2 & SAM | | **90.0** (± 0.66) | **89.5** (± 0.44) | **73.6** (± 0.15) | **77.3** (± 0.06) | **78.5** (± 0.73) | **70.1** (± 0.80) | **70.8** (± 1.01) | **77.6** (± 0.36) | **87.8** (± 0.47) | **75.7** (± 1.28) | **77.3** (± 1.28) | **74.4** (± 1.23) | **72.8** (± 1.09) |

Table 2: Results on shapes with small components of PartNet-E. The mIoU metric is measured in percentage.

| | Coffee Machine | | Door | | Laptop | | Lamp | | Safe | | Remote | | Overall (6) | |
|---|---|---|---|---|---|---|---|---|---|---|---|---|---|---|
| | Large | Small | Large | Small | Large | Small | Large | Small | Large | Small | Large | Small | Large | Small |
| PartSLIP [6] | 57.1 | 17.4 | 44.1 | 47.3 | 62.0 | 10.7 | 84.1 | 13.1 | 68.0 | 19.4 | - | 36.5 | 56.5 | 24.8 |
| PartSLIP++ [11] | 59.2 | 18.4 | 43.6 | 48.5 | 62.7 | 7.6 | 86.1 | 13.5 | 71.6 | 20.1 | - | 36.4 | 58.5 | 25.1 |
| PartSTAD [12] | 52.8 | 18.9 | 67.7 | 48.9 | 71.2 | 10.2 | 86.9 | 12.7 | 66.5 | 22.0 | - | 53.4 | 62.1 | 28.9 |
| Ours | 55.5 | 28.7 | 81.7 | 55.4 | 76.5 | 34.5 | 84.8 | 39.3 | 81.6 | 54.6 | - | 82.0 | 64.1 | 49.8 |

Table 3: Few-shot part segmentation results on ShapeNetPart. The mIoU metric is in percentage.

| Methods | VLM | Airplane | Cap | Car | Knife | Laptop | Mug | Pistol | Table | Overall(16) |
|---|---|---|---|---|---|---|---|---|---|---|
| PointCLIP [51] | CLIP | 27.2 | 61.1 | 30.2 | 72.2 | 89.9 | 59.6 | 55.6 | 53.0 | 52.1 |
| PointCLIPv2 [52] | CLIP | 37.1 | 66.8 | 38.3 | 73.3 | 89.1 | 75.7 | 68.4 | 56.3 | 57.0 |
| SegGraph (ours) | CLIP & SAM | **37.81** (± 0.73) | **73.96** (± 0.3) | **48.19** (± 1.52) | **74.8** (± 0.93) | **90.98** (± 0.27) | **80.08** (± 0.04) | **70.6** (± 1.16) | **72.72** (± 1.21) | **62.6** (± 1.00) |

and cameras (in Laptop). Both the quantitative results in the table and the qualitative visualizations in Fig. 5 indicate that our method exhibits clear advantages in accurately segmenting small parts.

In Fig. 5, we compare the predictions of SegGraph with those of previous approaches. Benefiting from the high-quality over-segmentation provided by SAM, our method achieves superior segmentation performance on small-sized parts compared to the 2D label aggregation methods used in the second and third rows. Notable examples include the hands of clocks, buttons on phones and remote controls, and handles on pot lids. Compared to the fourth row, which uses 3D feature-based method, our method exhibits greater advantages along the boundaries of small objects, such as the handle of a bucket and the clock hands. Overall, our approach yields better segmentation performance for small-size parts.

**Results on ShapeNetPart.** Tab. 3 presents the evaluation results on ShapeNetPart. Since only sparse point clouds without color are available for ShapeNetPart (2048 points), we follow PointCLIP-v2 to densify and smooth point clouds in preprocessing. Therefore, the rendering depth images of ShapeNetPart are of significantly lower quality compared to the rendered images of PartNet-E, leading to worse SAM segmentation and pretrained image features. However, our method still prevails against baseline methods with a $5.6\%$ improvement in mIoU.

## 4.2 Ablation Studies

We ablate each module in our method. As the vanilla baseline, we aggregate multi-view 2D features extracted by DINOv2 [43] onto the 3D space using average pooling, resulting in 3D feature representation. A MLP is then employed as a classifier to perform point-wise segmentation on the aggregated 3D features without using SAM segments. The performance of this vanilla baseline is shown in the first row of Tab. 4. All experiments are conducted on all shape categories of PartNet-E with an 8-shot setting. We leave hyperparameter analysis and more results to the supplementary.

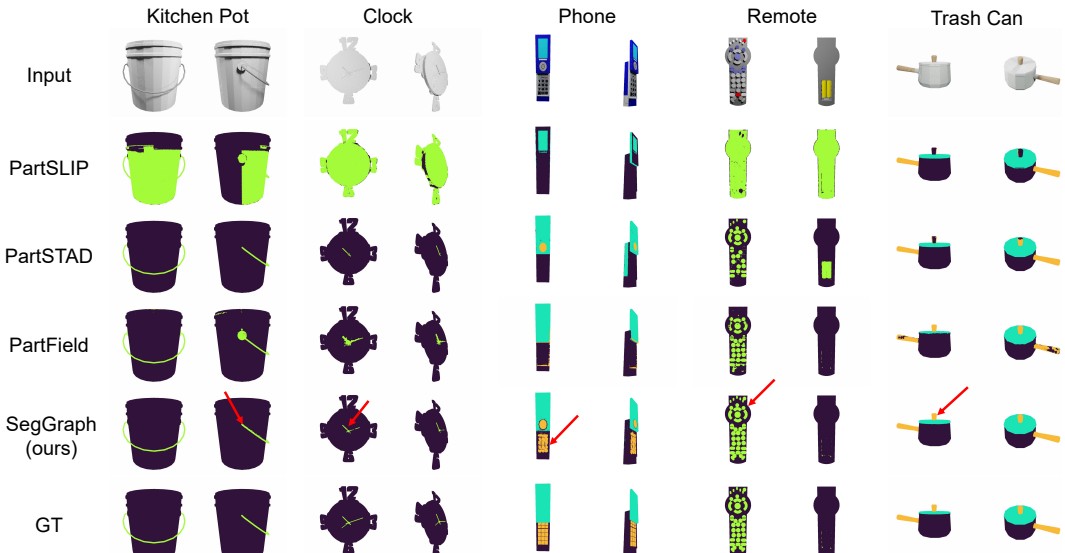

Figure 5: Qualitative comparison of part segmentation results on the PartNet-E dataset under the few-shot setting.

Table 4: Ablation experiments. SE is for segment encoding and AVE for mean pooling or unpooling. SAM means using 3D segments from SAM segmentation.

| | SAM | AVE→SE | AVE→Eq.(2) | $\mathcal{E}_o$ | $\mathcal{E}_a$ | Bottle | Door | Suitca. | Overall(45) |
|---|---|---|---|---|---|---|---|---|---|
| (1) | | | | | | 83.8 | 62.2 | 70.3 | 65.2 |
| (2) | ✓ | | | | | 89.0 | 72.7 | 70.5 | 70.5 |
| (3) | ✓ | ✓ | | | | 88.7 | 72.8 | 67.7 | 70.7 |
| (4) | ✓ | | ✓ | | | 88.7 | 72.5 | 65.5 | 70.5 |
| (5) | ✓ | ✓ | ✓ | | | 87.9 | 72.3 | 66.9 | 71.0 |
| (6) | ✓ | | | ✓ | ✓ | 87.8 | 70.5 | 73.0 | 71.4 |
| (7) | ✓ | ✓ | ✓ | ✓ | | 89.0 | 73.3 | 74.4 | 72.2 |
| (8) | ✓ | ✓ | ✓ | | ✓ | 89.7 | 72.6 | 74.0 | 72.3 |
| (9) | ✓ | ✓ | ✓ | ✓ | ✓ | 90.0 | 73.6 | 77.3 | 72.8 |

Table 5: Results of using different foundation models and feature propagation (SegGraph vs MLP).

| | | Bottle | Door | Suitca. | Overall (45) |
|---|---|---|---|---|---|
| CLIP [42] | MLP | 63.2 | 35.0 | 44.0 | 39.3 |
| | SegGraph | 71.2 | 36.5 | 52.1 | 49.5 |
| Diffusion [9] | MLP | 78.8 | 54.0 | 59.1 | 56.8 |
| | SegGraph | 85.0 | 59.0 | 69.4 | 64.3 |
| GLIP [8] | MLP | 82.1 | 57.7 | 62.1 | 61.1 |
| | SegGraph | 87.4 | 69.5 | 72.8 | 70.4 |
| PartField [15] | MLP | 75.9 | 65.4 | 72.0 | 66.3 |
| | SegGraph | 76.6 | 70.9 | 75.9 | 70.3 |
| DINOv2 [43] | MLP | 82.0 | 62.8 | 68.5 | 65.2 |
| | SegGraph | 90.0 | 73.6 | 77.3 | 72.8 |

**The Role of Segment Encoding (SE) and Viewing Quality-Awareness.** The settings without check marks for *AVE→SE* use average pooling, while those without check marks for *AVE→Eq.(2)* removes viewing quality awareness and uses average unpooling, instead. In rows 3-4 of Tab. 4, applying either SE or Eq.(2) individually results in only marginal or even no improvement. However, using both modules together (row 5 vs row 2) leads to slight improvement (0.5%).

**The Role of Different Segment Relations.** In rows 2,6 of Tab. 4, even without segment encoding and viewing quality awareness, incorporating segment graph leads to segment improvement (+0.9%), suggesting the positive role of graph propagation with GATv2. We conjecture these two graph edges encode geometric features and segment relations, leading to more consistent part segmentation.

Comparing rows 5,7,8 of Tab. 4, using either kind of graph edges for feature propagation with segment encoding and viewing quality awareness exhibits significant performance improvements (1.2% for $\mathcal{E}_a$ and 1.3% for $\mathcal{E}_o$, respectively). Using both edges simultaneously lead to a performance gain of 1.8% (see rows 5,9).

**Robustness to Different Foundation Model Features.** We conducted experiments by replacing DINOv2 [43] with pretrained features of three different foundation models: CLIP [42], Diffusion [9], GLIP [8], and PartField [15]. Among them, PartField is a 3D model distilled from a 2D foundation model. Since most comparison methods in Table 1 adopt GLIP, for fair comparison, we also modified GLIP into a feature extractor for our experiments. For GLIP, we utilized its multi-scale visual features fused with text features, and further applied a Feature Pyramid Network (FPN) structure to integrate these multi-scale representations, thereby transforming GLIP into a feature extractor. As a comparison, we also replaced the graph propagation module with a MLP to assess the role of structural

Table 6: Detailed runtime analysis of each component in the data preprocessing stage in our method compared with PartSLIP++. All runtime is measured on a per-shape basis.

| Method | Render (s) | GLIP+SAM / SAM (s) | Voting Preprocess / Build Graph (s) | Total (s) |
|---|---|---|---|---|
| PartSLIP++ | 2.12 | 16.33 (GLIP+SAM) | 92.38 (Voting Preprocess) | 110.30 |
| Ours | 2.12 | 52.40 (SAM) | 10.40 (Build Graph) | 64.92 |
| Ours | 2.12 | 1.30 (FastSAM) | 10.40 (Build Graph) | 13.82 |

Table 7: Training and inference time comparison per shape.

| Method | Train (s) | Inference (s) |
|---|---|---|
| PartSLIP++ | — | 5.83 |
| Ours | 2.25 | 1.46 |

reasoning. As shown in Tab. 5, when replacing the MLP with our proposed graph propagation module, we observe consistent and significant performance improvements across all foundation models. This demonstrates that our method is not only well-suited for 2D-to-3D knowledge transfer, but also effectively enhances the performance of 3D segmentation models. These results validate the strong generalizability of our approach across different types of feature representations.

## 4.3 Runtime Analysis

Tab. 6 and Tab. 7 present the runtime analysis of each component and comparison with PartSLIP++ on the PartNet-E dataset. All measurements are on a per-shape basis, with training time referring to one epoch for a single shape, using a single NVIDIA V100 GPU.

For PartSLIP++, most preprocessing time is spent on superpoint generation via L0-cut pursuit and voting weight computation (92.38 s). Its weighted voting during inference also results in high latency (5.8 s).

For our method, the main overhead stems from SAM segmentation on ten multi-view images. As the SamAutomaticMaskGenerator processes one image at a time, this step is relatively slow but still faster than PartSLIP++ (64.9 s vs. 110.8 s). Employing faster variants such as FastSAM [54] further reduces the time but slightly degrades accuracy (72.8 → 70.9 mIoU).

During inference, our method is more efficient (1.46 s vs. 5.83 s). Since PartSLIP++'s training code is unavailable, its training time is omitted. However, we can reasonably expect that our training time is much faster than PartSLIP++'s based on the inference time comparison. Note that both the "Train" and "Inference" time measurements include the time for DINOv2 feature encoding.

## 4.4 More Analysis

**Feature Visualization**    To more clearly demonstrate the capability of our method in learning 3D structural information, we visualize extracted 3D features in Fig. 6. As a baseline for comparison, we use 3D features obtained by directly averaging DINOv2 [43] features of multi-view rendered images, without any graph-based propagation. We apply PCA to reduce the feature channels to 3, corresponding to the RGB color channels for visualization. Fig. 6 suggests that the features processed by SegGraph exhibit clear separability among different parts. This strong discriminative ability arises from two aspects. First, the few-shot fine-tuning process enables the model to learn meaningful part-level distinctions even the number of shots is quite low (8-shot). Second, it benefits from the proposed graph design to encode 3D structural information. For instance, regions such as the screen area of the phone and the supporting bracket at the top of the coffee machine on the far right-both of which lack annotations in the training set—still exhibit clear part-level separability in the feature visualization.

**Feature Similarities Across Shapes.**    Given the strong part-level separability observed in Fig. 6, we further explore feature similarities in Fig. 7. Specifically, after selecting an anchor point within a shape (indicated by the red arrow), we compute the Euclidean distance between the feature of each point and that of the anchor point, either within the same shape or across different shapes (as shown in the second row). The distances are then normalized, with a distance of 0 mapped to blue and a distance of 1 mapped to gray. As demonstrated in the figure, the selected anchor point yields consistent part-level feature similarity, both within the same shape and across different shapes.

**Failure Cases.**    We examined the visual results and observed an interesting failure case in the USB category (Fig. 8). The USB model contains two disconnected parts—the cap and the body—each with a visually and geometrically similar connection subpart. While SAM correctly distinguishes these subparts, DINOv2 features for them are nearly identical despite their spatial separation. Consequently,

SegGraph misclassifies the two as belonging to the same category. This case indicates that SegGraph can be confused by repeated or similar sub-parts even when they are spatially distant.

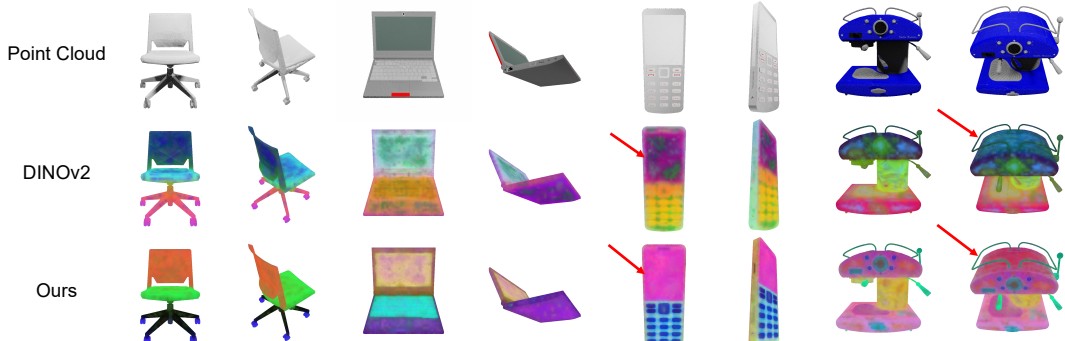

Figure 6: Feature Visualization. Points with similar colors are likely to share the same part label.

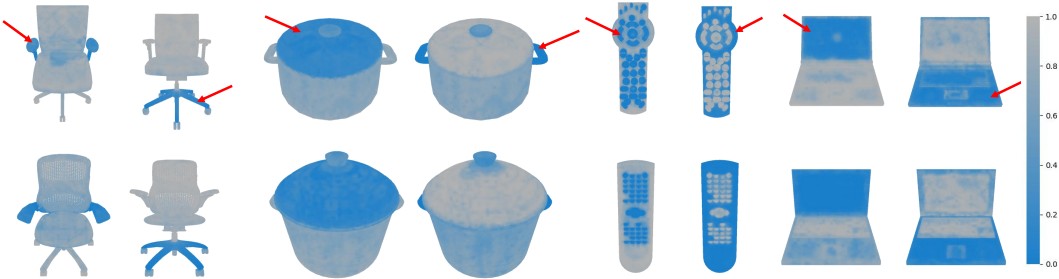

Figure 7: Feature similarities between an anchor point (red arrows) and points in the same shape (top row) and points in a different shape (bottom row). We use colors to encode feature similarities.

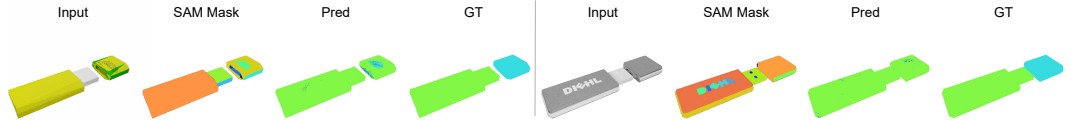

Figure 8: Failure cases on the USB category

## 5 Conclusion and Discussion

We presented SegGraph, a SAM segment-based graph propagation method for utilizing 2D features of foundation models for 3D part segmentation. Based on multi-view segmentation masks generated by SAM, we construct a graph to encode geometric features with segments as nodes and spatial relationships between nodes as edges. Through feature propagation on this graph, our method effectively adapts features from 2D foundation models to the 3D domain. SegGraph achieves state-of-the-art performance on 3D part semantic segmentation, especially for part boundaries and small components.

While our method demonstrates strong performance, there are some limitations. First, the rendering-view based feature aggregation paradigm fundamentally limits our ability to handle occluded or internal structures of 3D models. Second, the current framework operates at a fixed segmentation scale, without accommodating multi-scale part granularity. Future work could explore constructing hierarchical semantic representations of shapes, facilitating more versatile applications.

**Acknowledgement.** We thank all the anonymous reviewers for their insightful comments. This work was partially supported by the National Natural Science Foundation of China (62271467, 62476262, 62206263, 62306297, 62306296), Beijing Nova Program, and Beijing Natural Science Foundation (4242053, L242096).

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
