# OpenReview forum: "SegGraph: Leveraging Graphs of SAM Segments for Few-Shot 3D Part Segmentation"
_NeurIPS.cc/2025/Conference — NeurIPS 2025 poster_

### Official Review · Reviewer_hPPy · 2025-06-24

**Clarity:** 3
**Significance:** 3
**Originality:** 3
**Rating:** 4
**Confidence:** 4

**Summary:**

This paper presents a segment graph for few-shot 3D part segmentation. It encodes 3D point features through multi-view feature pooling from 2D foundation model outputs. Simultaneously, they generate view-consistent 3D segments by aggregating SAM-based 2D segmentations across all views. They then model both the spatial relationships (overlapping/adjacent) between segments and the constituent relationships between points and segments. The 3D point features are combined with segment features, which are sent to MLP for part prediction.

**Questions:**

1. Is the graph constructed from the segments of multi-views? How large is the graph? How many nodes and edges?
2. How to compute point normals at eqn 2?

**Ethical Concerns:**

["NO or VERY MINOR ethics concerns only"]

**Final Justification:**

The rebuttal is good. I keep my rating positive.

**Limitations:**

see weakness.

**Quality:**

3

**Strengths And Weaknesses:**

Strength:
1. figure 1 clearly show the difference between theirs and existing methods. The segment graph is novel to me.
2. achive sota performance.
3. extensive experiments.

weakness:
1. Some part is not clear to me, such as line 148-153. There are several different aggregation methods. Can you explain why?
2. I guess the optimization of the graph is time consuming. Can you provide a optimization time comparison with other models?

---

> ### Author Rebuttal · Authors · 2025-07-31
>
> Thank you for your valuable comments and kind words to our work. Below we address specific questions.
>
> **Q1: Some part is not clear to me, such as line 148-153. There are several different aggregation methods. Can you explain why?**
> **ANS:** The use of different aggregation methods in lines 148–153 is intentional and serves distinct yet complementary purposes. The first aggregation (line 150) applies max pooling to encode segment features. However, simple max-pooling ignores the varying contributions of different points within a segment. For example, points close to boundaries are usually more inconsistent with the segment label and should be downweighted.
>
> To address this problem, the second aggregation (line 152) refines the segment representation by performing a weighted aggregation of point-level DINOv2 features. This weighting mechanism leverages the pooled segment feature to estimate the importance of each point, allowing the model to downweight noisy or ambiguous points—particularly those near segment boundaries that may partially belong to other categories. This design enhances the robustness of segment-level semantic features by mitigating boundary noises and improving within-segment consistency.
>
> **Q2: I guess the optimization of the graph is time consuming. Can you provide a optimization time comparison with other models?**
> **ANS**： The optimization of the graph is quite efficient. In the table below, we report the optimization time for one epoch, the inference time for a single shape, and the inference time of PartSLIP++. Our method is more efficient than PartSLIP++. Since the training code of PartSLIP++ is not publicly available, we do not report its training optimization time. Nevertheless, the one epoch training time required for our graph optimization is still smaller than the inference time of PartSLIP++.
> The preprocess time for the two methods:
>
> | Method     | Render (s) | GLIP+SAM / SAM (s) | Voting Preprocess / Build Graph (s) | Total (s) |
> | :--------- | :--------- | :----------------- | :---------------------------------- | :-------- |
> | PartSLIP++ | 2.12       | 16.33 (GLIP+SAM)   | 92.38 (Voting Preprocess)           | 110.30    |
> | Ours       | 2.12       | 52.40 (SAM)        | 10.40 (Build Graph)                 | 64.92     |
>
> The training and inference time:
>
> | Method     | Training (s) | Inference (s) |
> | :--------- | :----------- | :------------ |
> | PartSLIP++ | —            | 5.83          |
> | Ours       | 2.25         | 1.46          |
>
> **Q3: Is the graph constructed from the segments of multi-views? How large is the graph? How many nodes and edges?**
> **ANS**: Yes, the nodes in our graph are directly formed from the segments obtained by applying SAM to multi-view images. To quantitatively illustrate the size of the constructed graphs, we report the average and maximum number of nodes, adjacent edges, and overlapping edges per object on the PartNetE dataset.
>
> |      | Nodes  | AdjEdges | OvlpEdges |
> | :--- | :----- | :------- | :-------- |
> | mean | 136.53 | 310.58   | 2386.84   |
> | max  | 1121   | 1872     | 21024     |
>
> As shown in the table, the constructed graph (with an average of a few hundred nodes and several thousand edges) is significantly smaller compared to the point clouds of each object in PartNetE, which typically contain hundreds of thousands of points.
>
> **Q4: How to compute point normals at eqn 2?**
> **ANS**： We estimate the point cloud normals as follows. For each point, we consider a local neighborhood within a radius of 0.02 and employ the built-in function in Open3D to estimate the normals. Specifically, Open3D performs a PCA analysis on the local neighborhood of each point, and the normal vector is taken as the eigenvector corresponding to the smallest eigenvalue of the covariance matrix. Optionally, Open3D applies normal orientation adjustment to ensure consistency in the normal directions.
>
> **If you have any further question, please let us know. Thank you very much!**

---

> > ### Comment · Reviewer_hPPy · 2025-08-06
> >
> > The rebuttal is good. I keep my rating positive.

---

> > > ### Author Response · Authors · 2025-08-06
> > >
> > > Thank you for your positive evaluation and supportive comments.

---

### Official Review · Reviewer_epFc · 2025-07-01

**Clarity:** 4
**Significance:** 2
**Originality:** 3
**Rating:** 5
**Confidence:** 5

**Summary:**

The paper tackles the task of few-shot 3D part segmentation through the approach of 2D-3D knowledge transfer. The paper proposes SegGraph. It constructs a graph, with nodes as 3D segments and edges as 3D segments spatial relationship (overlap or adjacency). It refines the back-projected DINOv2 feature through a Graph Attention Network and conducts "viewing quality-aware feature unpooling" to reweigh features from each view. The idea is to break SAM segmented masks into independent pieces and reconnect them through a graph structure. Experiments show that the number and visual quality outperform the SOTA.

**Questions:**

* The method was mainly evaluated on clean datasets with reliable geometry and colors. How does the method perform for real-world data? What about its performance on raw and noisy point clouds from real world. For example, the algorithm reconstructed 3D objects, 3D assets represented by 3DGS.  Can the graph be correctly built and maintain the performance?

* Concerns of performance contributions. See weakness #2

* How does the method compare with semantic mask-based approaches? They may not focus on the same task in their papers, but exhibit capabilities in SegGraph's tackled task. Some of them even have performance advantages.

Refs:
[1] OpenMask3D: Open-Vocabulary 3D Instance Segmentation. NeurIPS 2023. [2] SATR: Zero-Shot Semantic Segmentation of 3D Shapes. ICCV 2023. [3] Open-Vocabulary Semantic Part Segmentation of 3D Human. 3DV 2025. [4] Find Any Part in 3D. Arxiv in 2024.

**Ethical Concerns:**

["NO or VERY MINOR ethics concerns only"]

**Final Justification:**

The rebuttal addresses most of my concerns. Please incorporate these additional evaluations (& anything incomplete due to time constraint of rebuttal) in the final version.

**Limitations:**

yes

**Quality:**

3

**Strengths And Weaknesses:**

Strengths
* The paper has valid motivation: to improve the 2D-3D knowledge transfer.
* Clear writing. The method is easy to follow, and the experiments done have comprehensive results provided.
* Visualization looks promising. The experiment table shows performance improvements over SOTA, such as Partslip and Partslip++.

Weaknesses
* As stated, the granularity problem. Currently, it operates at a fixed granularity scale. The dataset is also labeled on a consistent scale. If the few-shot examples are under various granularity scales, the performance of the graph formulation is questionable.
* The overall performance increase may be attributed to the application of powerful DINOv2 rather than the proposed method. Tab.5 shows the efficacy of SegGraph, but also shows that when replacing it with CLIP, the SegGraph only has 49.5 mIOU, lower than most of GLIP+SAM methods in Tab.1, which is basically CLIP+SAM.
* The proposed method seems not be able to handle the semantically different but geometrically similar parts.

---

> ### Author Rebuttal · Authors · 2025-07-31
>
> Thank you for your valuable comments and kind words to our work. Below we address specific questions.
>
> **Q1： If the few-shot examples are under various granularity scales, the performance of the graph formulation is questionable.**
> **ANS:** Our method can be extended to multi-granularity once the annotation is available. For example, 3DComPat++ \[1\]  provides coarse-grained annotations (at most 5 part categories) and fine-grained annotations (10-20 part categories). Due to limitations in time and computational resources, we selected three categories—**airplane**, **bicycle**, and **chair**—for evaluation. Airplanes have 14 fine-grained and 3 coarse part labels; bicycles have 26 fine-grained ones and 3 coarse ones; chairs have 22 fine-grained labels and 4 coarse labels.
> We extend to multi-grained segmentation by tuning separate networks. In the table below, our method demonstrates significant advantages across different levels of part annotation granularity.
>
> |                       |  Coarse  | Coarse  | Coarse |   Fine   |  Fine   | Fine  |
> | :------ | :------ | :------ | :------ | :------ | :------ | :------ |
> |                       | airplane | bicycle | chair  | airplane | bicycle | chair |
> | PartSLIP++(zero-shot) |  11.56   |  60.52  | 16.59  |   9.48   |  2.60   | 4.87  |
> | Ours(1-shot)          |  93.31   |  91.21  | 41.41  |  45.34   |  26.26  | 27.21 |
> | Ours(8-shot)          |  94.64   |  95.46  | 69.26  |  65.38   |  54.17  | 42.61 |
>
> **Q2： The overall performance increase may be attributed to the application of powerful DINOv2 rather than the proposed method. Tab.5 shows the efficacy of SegGraph, but also shows that when replacing it with CLIP, the SegGraph only has 49.5 mIOU, lower than most of GLIP+SAM methods in Tab.1, which is basically CLIP+SAM.**
> **ANS:** We would like to point out that GLIP is generally more powerful than CLIP. CLIP primarily focuses on global semantic features at the image level and falls short in localization tasks, such as detections and part segmentation, as illustrated in Figure 5 of the paper "*Foundational Models Defining a New Era in Vision: A Survey and Outlook".* In contrast, GLIP is designed for object detection and is trained using large-scale detection datasets. Therefore, GLIP demonstrates advantages in fine-grained localization and facilitates much better part localization than CLIP, leading the performance gaps when using different backbone features.
>
> To isolate the influence of foundation models, we evaluate the results of using GLIP as the 2D feature extractor in place of DINOv2 on the PartNetE dataset. Specifically, we used the multi-scale visual features from GLIP after fusion with text features, and employed a Feature Pyramid Network (FPN) structure to further fuse these multi-scale features. The resulting fused features were then used as the GLIP features for SegGraph aggregation. However, we need to notice that this design is inferior as it ignores the bounding box outputs. The results in the table below show that even when using GLIP, our method still achieves competitive performance, second only to our results using DINOv2. This demonstrates that the overall performance increase is not solely attributed to the application of the powerful DINOv2.
>
> | Method | PartSLIP | PartSLIP++ | PartSTAD | PartDistill | 3-By-2 | Ours-DINOv2 | Ours-GLIP |
> | :----- | :--------- | :--------- | :--------- | :--------- | :--------- | :--------- | :--------- |
> | mIoU   | 59.4     | 62.1       | 65.0     | 65.9        | 64.2   | 72.8        | 70.4      |
>
> **Q3： The proposed method seems not be able to handle the semantically different but geometrically similar parts.**
> **ANS**： Thanks to the semantically rich features extracted from DINOv2, our method is capable of segmenting parts that share similar geometric structures but differ in semantic meaning. For example, the screen and the base of a laptop are both rectangular and flat in shape, exhibiting high similarity in 3D geometry. However, as shown in Figure 6, visualizations of their features reveal that they are highly distinguishable in the feature space.
>
> **Q4：How does the method perform for real-world data? What about its performance on raw and noisy point clouds from real world. For example, the algorithm reconstructed 3D objects, 3D assets represented by 3DGS. Can the graph be correctly built and maintain the performance?**
> **ANS**： To verify the robustness of our method to noise, we conducted experiments by adding noise in the normalized point cloud space. We evaluate the method against added noises with a standard deviation of {0.005, 0.01, 0.02，0.05}. Extending the method to 3DGS requires substantially changing the pipeline, and we leave it to future work.
>
> |            |  0   | 0.005 | 0.01  | 0.02  | 0.05  |
> | :--------- | :--- | :--- | :--- | :--- | :--- |
> | KitchenPot | 85.2 | 81.83 | 80.29 | 75.01 | 66.68 |
> | Bottle     | 90.0 | 83.98 | 78.99 | 74.88 | 69.04 |
> | Chair      | 89.5 | 87.03 | 82.34 | 78.02 | 67.58 |
>
> **Q5：How does the method compare with semantic mask-based approaches? They may not focus on the same task in their papers, but exhibit capabilities in SegGraph's tackled task. Some of them even have performance advantages.**
> **ANS：** The following table provides a comparison with the methods you mentioned—OpenMask3D, SATR, and Find Any Part in 3D (Find3D). The method "Open-Vocabulary Semantic Part Segmentation of 3D Human" is not included in the comparison because its code is not publicly available. Among these, our approach consistently demonstrates a significant performance advantage. One important factor is that OpenMask3D relies on Mask3D as its 3D feature extractor, which is pre-trained on ScanNet—an indoor scene dataset. This domain gap makes it difficult for OpenMask3D to generalize well to the more diverse and structurally complex 3D shapes in PartNetE. Moreover, as mentioned in the Find3D paper, PartNetE is particularly challenging due to the presence of many small parts that lack distinct geometric or color features (e.g., buttons on a surface of the same color), which significantly hinders the performance of Find3D and SATR. In contrast, our method excels in identifying and segmenting such small, indistinct components, which underscores its robustness in fine-grained part segmentation.
>
> | Setting   | Method         | mIoU   |
> |:---------|:--------------|:------|
> | Zero-shot | OpenMask3D     | 12.54  |
> | Zero-shot | Find3D         | 16.38  |
> | Zero-shot | SATR           | 19.42  |
> | Few-shot  | ours (1-shot)  | 53.36  |
> | Few-shot  | ours (8-shots) | 72.80  |
>
>
> \[1\] 3DCoMPaT^{++}: An improved Large-scale 3D Vision Dataset for Compositional Recognition
>
> **If you have any further question, please let us know. Thank you very much!**

---

> ### Comment · Reviewer_epFc · 2025-08-03
>
> Thanks for the response. It addresses most of my points.
>
> Regarding the experiment for multi-granularity labels, you mentioned: "We extend to multi-grained segmentation by tuning separate networks." Does it mean you have a network for coarse and a network for fine? My original point is that usually, the granularity of real data label is mixed. It's difficult to just classify them as "coarse" or "fine". Normally, the granularity forms a continuous spectrum. It may not be realistic to tune a bunch of networks. How can this graph structure handle examples under various granularities? That's my original question.

---

> > ### Author Response · Authors · 2025-08-04
> >
> > We sincerely thank you for your efforts in reviewing the rebuttal and the main paper.
> >
> > Yes, we trained and evaluated separate networks on datasets with coarse-grained and fine-grained annotations, respectively. This design assumes a consistent and fixed part hierarchy level across shapes within the same category. Under this assumption, labels at each level of granularity can be handled by a separate network with a shared backbone and separate classification heads. However, as you rightly pointed out, in real-world scenarios, part granularity may have diverse hierarchy definitions, and there is no clear granularity level (i.e., “a continuous spectrum of the part granularity” and “mixed granularity”), making it infeasible to train a dedicated classification head for every possible level. This highlights a limitation of our method when applied to real-world data with diverse granularities, as acknowledged in the limitations of the main paper.
> >
> > Regarding the graph-based aggregation (i.e., “graph structure”), it is independent of the issue of mixed-granularity part segmentation.
> >
> > If there are any misunderstandings of your comments or unclear parts, please let us know.

---

### Official Review · Reviewer_v5oZ · 2025-07-02

**Clarity:** 2
**Significance:** 3
**Originality:** 3
**Rating:** 5
**Confidence:** 4

**Summary:**

This paper proposes SegGraph, a method for few-shot 3D part segmentation that leverages 2D foundation models and a graph-based segmentation module. The approach begins by rendering multi-view images of a 3D point cloud shape and extracting segmentation masks using SAM, along with visual features from DINOv2. The 2D segments are collected across all rendered views from SAM. An embedding is learnt using the local and semantic features, which forms nodes in a segment graph. The graph captures both overlapping and adjacent relationships between segments and is processed using a GATv2 network. These features are then reweighted via a view quality-aware mechanism and unpooled back to individual 3D points for final part segmentation. The model is trained in a few-shot setting and evaluated on two benchmark datasets, demonstrating strong performance, especially on small parts.

**Questions:**

Questions (considered for score change):
1. Baseline and foundation model justification: Could the authors provide results or justification for using consistent baselines across all datasets, and for isolating gains from the choice of foundation model?
2.  Significance of thresholds used for graph edges: The paper defines overlapping segments as those with mIoU greater than 10%, and adjacent segments as those whose closest points are less than 0.01 units apart in normalized space. Could the authors justify these thresholds and provide an analysis of how sensitive the model is to these values?

Clarifying questions:
1. What is the number of shapes per category in the few-shot experiments? Reporting the range (e.g., min/max examples per class) would help interpret the model's generalization performance across both frequent and rare classes.
2. The paper highlights significant improvements on small parts, but does not define how "small" is quantified. Are small parts determined based on point count, spatial volume, or proportion of the object?
3. Clarification of “w/o SAM” setting in ablation: In Table 4, the model is evaluated without SAM. How are 3D segments generated in this setting, and how is the pipeline modified to function without SAM-derived masks?

**Ethical Concerns:**

["NO or VERY MINOR ethics concerns only"]

**Final Justification:**

The paper provides valuable contribution in the setting of 3D part segmentation. It is well written, it covers recent baselines and provides a thorough comparison. During the initial review there were concerns about where the gains in the pipeline were coming from, inadequate discussion on the failure cases / setting with complex shapes, and the time cost analysis. The authors engaged during the rebuttal to clarify these concerns and therefore, I have increased my rating to Accept.

**Limitations:**

While the method performs well overall, there is little to no qualitative or quantitative discussion of where it fails (e.g., occluded parts, inconsistent multi-view alignment).

**Quality:**

3

**Strengths And Weaknesses:**

### Strengths

1. Clarity: The paper is well-written and clearly structured. It presents a well-motivated approach, thoroughly discusses prior work, and explains its contributions clearly.
2. Empirical results: The method demonstrates strong performance in the few-shot 3D part segmentation setting on both PartNet-Ensemble and ShapeNetPart datasets.
3. View Quality-Aware Unpooling: It is a novel contribution and can be useful for other tasks as well.
4. Robustness of results: The authors report mean and standard deviation over multiple training runs, highlighting the method’s stability and reproducibility. It's a good experimental practice.

### Weaknesses:

1. Ambiguity in key definitions and notation: The paper occasionally introduces parameters without a formal definition. For example, the mathematical representation of segment nodes $V$ and $F^{p^{'}}_i$ used in equation 3. Clearer definitions would improve readability. Adding equations to explain L148-153 mathematically will also improve clarity.

2. Certain technical statements are imprecise. For instance, SATR does not use a voting-based aggregation scheme as implied in L31–34. Similarly, referring PartField as a zero-shot method (L205) is confusing as it requires part-level supervision during training.

3. Writing clarity and specificity: Some claims lack necessary specificity. For example, the phrase “an improvement of at least 4%” in L69 does not clarify the metric, baseline, or dataset. The abstract also omits the fact that the method operates on 3D point clouds. Additionally, Equation 1 would benefit from a clarifying sentence noting that the denominator is computed per axis and that the normalisation is element-wise.

4. Baseline and choice of foundation model: Different baselines are reported on different datasets. The authors can either explain these omissions (for the models which have public code that can be applied to untextured shapes) or provide consistent baselines across datasets to allow for fairer comparisons. In Table 1, different methods use different foundation models (e.g., GLIP, DINOv2, SAM), which complicates the interpretation of performance gains. If feasible, providing comparisons with a fixed foundation model would more clearly isolate the contribution of the proposed 3D pipeline.

[Optional] Figure 2: The figure is general and does not precisely capture the complexity of the pipeline.

[Minor] Line 59: “all views” is ambiguous—suggest replacing with “all training views,” as the number of possible renderings is theoretically unlimited. Figure 3 shows adjacency (solid lines) only within the same view, but the text does not clarify if the adjacency edges can exist across views as well.

---

> ### Author Rebuttal · Authors · 2025-07-31
>
> Thank you for your valuable comments and kind words to our work. Below we address specific questions.
>
> **Q1: Baseline and foundation model justification: Could the authors provide results or justification for using consistent baselines across all datasets, and for isolating gains from the choice of foundation model?**
>
> **ANS:** We select DINOv2 as the base foundation model due to its good generalization and high-quality output features, achieved through large-scale self-supervised learning. Most baseline methods rely on foundation models with 2D label output, e.g., GLIP, and do not support the use of DINOv2. Moreover, the training codes of  PartSLIP, PartSLIP++, PartSTAD, 3-By-2, and PartDistill are not fully available, making it difficult to change their base foundation model for alignment.
>
> To isolate the influence of foundation models, we evaluate the results of using GLIP as the 2D feature extractor in place of DINOv2 for our method. Specifically, we used the multi-scale visual features from GLIP after fusion with text features, and employed a Feature Pyramid Network (FPN) structure to further fuse these multi-scale features. The resulting fused features were then used as the GLIP features for SegGraph aggregation. However, we need to notice that this design is inferior as it ignores the most important bounding box outputs. The results in the table below show that even when using GLIP, our method still achieves competitive performance, second only to our results using DINOv2.
>
> | Method | PartSLIP | PartSLIP++ | PartSTAD | PartDistill | 3-By-2 | Ours-DINOv2 | Ours-GLIP |
> | :----- | :------- | :--------- | :------- | :---------- | :----- | :---------- | :-------- |
> | mIoU   | 59.4     | 62.1       | 65.0     | 65.9        | 64.2   | 72.8        | 70.4      |
>
> **Q2: Significance of thresholds used for graph edges**
> **ANS:**  We validated the influence of these thresholds in Figures A2 and A3 in the supplementary material. Basically, the method is quite robust to these hyperparameters owing to the good quality of SAM segments.
>
> **Q3: What is the number of shapes per category in the few-shot experiments? Reporting the range (e.g., min/max examples per class) would help interpret the model's generalization performance across both frequent and rare classes.**
> **ANS:** Thank you for pointing out the omission regarding the number of shots used in our experiments. All experiments in the paper were conducted using an 8-shot per shape configuration. Additionally, we provide an analysis of the model’s generalization performance under different numbers of shots in Appendix C and Figure A1.
>
> **Q4：The paper highlights significant improvements on small parts, but does not define how "small" is quantified. Are small parts determined based on point count, spatial volume, or proportion of the object?**
> **ANS:** For the categorization of "small parts," we follow the approach used in PartSTAD, which defines small parts based on empirical knowledge of object part sizes. In the PartNetE dataset, the following categories are considered small parts: \["handle", "button", "wheel", "knob", "switch", "bulb", "shaft", "touchpad", "camera", "screw", "handlebar", "trigger"\].
>
> **Q5: Clarification of “w/o SAM” setting in ablation: In Table 4?**
> **ANS:** We describe the configuration without using SAM in Lines 253–257. In this setting, we first aggregate multi-view features into the 3D space via average pooling to obtain $F^p$, and then directly apply an MLP to predict the category of each point. The graph structure and SAM segments are completely discarded in this configuration.
>
> **Q6: While the method performs well overall, there is little to no qualitative or quantitative discussion of where it fails (e.g., occluded parts, inconsistent multi-view alignment).**
>
> **ANS:** We expect the method will fail on shapes with deep concave parts and inner structures because of occlusions. We acknowledge this is a fundamental limitation of multi-view rendering-based methods.
>
> **Q7: Writing and clarity**
> **ANS:** Thank you for your valuable suggestions regarding the writing and clarity of the paper. We will revise the relevant sections to improve the overall clarity. More explanations are as follows.
>
> 1. **SATR does not use a voting-based aggregation scheme as implied in L31–34. Similarly, referring PartField as a zero-shot method (L205) is confusing as it requires part-level supervision during training.**
>    The Per Face Score Aggregation module in SATR can be described as follows. Each 3D face receives initial votes from multiple 2D views via a 2D detector. These votes are then adjusted through a weighted refinement process based on 3D topology (geodesic distance) and neighborhood information (visibility). The final aggregated score determines the label of each face based on the highest vote score. Therefore, the overall approach still aligns with the description of "aggregating 2D labels to the 3D domain with voting".
>
>    Regarding PartField, we acknowledge that our original description was not fully accurate. PartField indeed requires training on data from Objaverse and PartNet with SAM-generated pseudo labels and contrastive losses. Though the method can work without ground truth annotations, referring to it as a zero-shot method is not entirely precise, as it still requires training.
> 2. **The phrase “an improvement of at least 4%” in L69 does not clarify the metric, baseline, or dataset:**
>    The reported "at least 4% improvement"  in L69 refers specifically to the performance on the PartNet-E dataset. It indicates that, under various configurations using different foundation models as image feature extractors, our method consistently yields at least a 4% improvement compared to not using SegGraph. We will actively integrate the suggestions into our revision.
> 3. **The mathematical representation of segment nodes $V$ and $F_i^{p'}$ in equation 3\. Adding equations to explain L148-153 mathematically will also improve clarity.**
>    Thank you. We will clarify these two parts.
> 4. **Figure 2: The figure is general and does not precisely capture the complexity of the pipeline.**
>    Thank you. We will incorporate the illustration for graph construction and view pooling in the revision.
> 5. **Figure 3 shows adjacency (solid lines) only within the same view, but the text does not clarify if the adjacency edges can exist across views as well.**
>    Thank you for the suggestion. We will add cross-view adjacency edges in Figure 2 to better illustrate the adjacency relationships across views.
>
> **If you have any further question, please let us know. Thank you very much!**

---

> > ### Comment · Reviewer_v5oZ · 2025-08-04
> >
> > Thank you to the authors for carefully responding to each comment. It addresses most of my concerns. I appreciate the experiment with GLIP, as it confirms that the observed performance improvements are not solely due to the choice of DINO.
> >
> > Regarding the explanation of SATR: I’d argue that describing SATR’s aggregation as “voting” can still be misleading. Traditional voting implies discrete, count-based decisions (e.g., majority vote), while SATR performs score aggregation and reweighting based on confidence, topology, and visibility. This distinction is important, especially given that one of SATR's key contribution lies in moving beyond simple vote-based label fusion. Considering L35-36 mention that "However, simply aggregating 2D labels to 3D domains overlooks the geometric structures of 3D inputs", and SATR addresses that limitation. So, I still think L31-34 should still provide additional clarification.
> >
> > Secondly, I still find the discussion on limitations and failure cases as somewhat generic. The stated limitations—such as occlusions in concave or internal structures—are common to a broad range of approaches. I was hoping to understand whether SegGraph, in particular, introduces new limitations or exhibits specific failure modes. Also, what is the granularity or complexity limit beyond which SegGraph fails to produce consistent segmentation.
> >
> > I am happy to increase my score to Accept if the authors can address the two remaining points above, and assuming the minor writing issues mentioned earlier and the time complexity analysis, as pointed by other reviewers, is addressed in the updated version.

---

> > > ### Author Response · Authors · 2025-08-05
> > >
> > > We sincerely thank you for your positive feedback and your devotion to the interactive discussions.
> > >
> > > We will revise the L31–34 accordingly to address the misleading voting description of SATR’s aggregation.
> > >
> > > Regarding more specific failure modes, we investigated the visual results and found an interesting failure case in the USB categories. In this case, the USB shape consists of two spatially disconnected components, that is, the cap and the body. Both the cap and the body have a corresponding connection subpart that is visually and geometrically similar. SAM can successfully identify these two distinct subparts. However, DINOv2 features for the two subparts are very similar, though the two subparts are geometrically separated. Therefore, SegGraph misclassifies these two subparts as the same category. This failure case suggests that SegGraph may be influenced by repeated or similar sub-parts, even when they are spatially distant and belong to different parts.
> > >
> > > We would like to express our gratitude for your constructive comments and will incorporate all suggested experiments and writing changes in the updated version.

---

> > > > ### Comment · Reviewer_v5oZ · 2025-08-06
> > > >
> > > > Thank you for providing additional details on the failure cases. It is helpful to know the limits of proposed method and I would suggest to include the failure cases in the updated version of the paper.
> > > >
> > > > The authors have adequately addressed all my concerns and I have increased the score.

---

> > > > > ### Author Response · Authors · 2025-08-06
> > > > >
> > > > > Thank you for your thoughtful feedback and for increasing the score. We will include the failure cases in the revised version.

---

### Official Review · Reviewer_URb2 · 2025-07-03

**Clarity:** 3
**Significance:** 3
**Originality:** 2
**Rating:** 4
**Confidence:** 4

**Summary:**

This paper presents SegGraph, a novel framework for few-shot 3D part segmentation that leverages SAM segments organized in a graph structure. The key innovation is constructing a segment graph where nodes represent 3D segments from multi-view SAM segmentation, and edges encode spatial relationships (overlap/adjacency). Features are propagated through this graph using GATv2, followed by view-quality-aware unpooling to map segment features back to 3D points. The method achieves state-of-the-art results on PartNet-E and ShapeNetPart datasets, showing particular strength in segmenting small parts and boundaries.

**Questions:**

1. How does the method perform on objects with significantly more complex part structures than those in PartNet-E?
2. What is the computational overhead compared to simpler baselines, particularly for inference?
3. How sensitive is the method to the number of views used for rendering?

**Ethical Concerns:**

["NO or VERY MINOR ethics concerns only"]

**Final Justification:**

The author addresses my questions, so I increase the score.

**Limitations:**

Yes

**Quality:**

2

**Strengths And Weaknesses:**

# Strengths
1. Well-motivated technical approach: The use of SAM segments as building blocks for 3D understanding is intuitive and well-justified. The observation that SAM provides high-quality grouping cues that can be leveraged for 3D part segmentation is valuable, and the graph-based propagation effectively captures geometric relationships.
2. Technical soundness: The segment graph construction with overlap and adjacency relationships is well-designed. The view-quality-aware unpooling module addresses the practical issue of varying SAM segment quality across viewpoints.
3. Clear presentation: The paper is well-written with clear motivation, good visual illustrations, and comprehensive experimental analysis.
# Weaknesses
1. Limited technical novelty: While the combination is effective, the individual components (multi-view rendering, SAM segmentation, graph neural networks) are not novel. The main contribution is in their combination, which limits the technical depth.
2. Computational overhead not adequately addressed: The method requires SAM inference on multiple views plus graph neural network computation. While Table 5 shows training time comparisons, inference time analysis is missing. The computational cost compared to simpler baselines should be more thoroughly analyzed.
3. Dependence on view quality: The method's performance is inherently limited by the quality of rendered views and SAM segmentation. While the view-quality-aware module partially addresses this, fundamental limitations remain for occluded or poorly-rendered regions.
4. Scalability concerns: The evaluation is limited to relatively small datasets and objects with moderate complexity. It's unclear how the method would scale to larger, more complex objects or datasets, like ScanNet [21], Objaverse [A], UDA-Part [B], or ABC [C] datasets.

[A] Deitke, Matt, et al. "Objaverse: A universe of annotated 3d objects." Proceedings of the IEEE/CVF conference on computer vision and pattern recognition. 2023.

[B] Liu, Qing, et al. "Learning part segmentation through unsupervised domain adaptation from synthetic vehicles." Proceedings of the IEEE/CVF conference on computer vision and pattern recognition. 2022.

[C] Koch, Sebastian, et al. "Abc: A big cad model dataset for geometric deep learning." Proceedings of the IEEE/CVF conference on computer vision and pattern recognition. 2019.

---

> ### Author Rebuttal · Authors · 2025-07-31
>
> Thank you for your valuable comments and kind words to our work. Below we address specific questions.
>
> **Q1: Limited technical novelty: While the combination is effective, the individual components (multi-view rendering, SAM segmentation, graph neural networks) are not novel. The main contribution is in their combination, which limits the technical depth.**
> **ANS:** As illustrated in Figure 1, the novelty of this work lies in the construction of a structured graph based on SAM segments as an efficient 3D representation to aggregate foundation model features. The graph effectively encodes the adjacency and overlapping relationships of SAM segments within a single view and across views. Moreover, this graph-based representation helps mitigate the label imbalance between small parts and large parts that have very different numbers of labeled points, by using a more uniform number of SAM segment nodes. For example, in the Laptop category, both the large screen and the small individual keys are represented as single nodes after SAM segmentation. Therefore, our method achieves better results on small parts.
> Besides, both you and Reviewer V5oZ acknowledge the novelty of View Quality-Aware Unpooling that is potentially beneficial for other tasks.
>
> **Q2:  The computational cost compared to simpler baselines should be more thoroughly analyzed.**
> **ANS:** The table below presents a detailed runtime analysis of each component in our method, alongside a comparison with the classical baseline PartSLIP++. All runtime is measured on a per-shape basis. The training time is counted for the optimization of one epoch of a shape.
> For PartSLIP++, the main computational cost during preprocessing lies in generating superpoints from the point cloud using L0-cut pursuit and computing voting weights for each superpoint (vote\_preprocess: 92.38 seconds). During inference, its complex weighted voting mechanism leads to a relatively high inference time (5.8 seconds).
> For our method, the main preprocessing cost arises from SAM segmentation for 10 multi-view images. Because the adopted SamAutomaticMaskGenerator processes only one image each time, this step is relatively time-consuming. However, faster models could be adopted in the future to accelerate this process. Even so, our preprocessing time is significantly lower than that of PartSLIP++ (64.921s vs. 110.83s).
> In the inference stage, our method achieves higher efficiency (1.46s vs 5.83s for PartSLIP++). It is worth noting that since the training code of PartSLIP++ is not publicly available, its training time is not reported in the table. However, we can reasonably expect that our training time is much faster than PartSLIP++'s based on the inference time comparison. Note that both the "Train" and "Inference" time measurements include the time for DINOv2 feature encoding.
>
> Time spent on preprocessing：
>
> | Method     | Render (s) | GLIP+SAM / SAM (s) | Voting Preprocess / Build Graph (s) | Total (s) |
> | :--------- | :--------- | :----------------- | :---------------------------------- | :-------- |
> | PartSLIP++ | 2.12       | 16.33 (GLIP+SAM)   | 92.38 (Voting Preprocess)           | 110.30    |
> | Ours       | 2.12       | 52.40 (SAM)        | 10.40 (Build Graph)                 | 64.92     |
>
> Time spent on training and Inference:
>
> | Method     | Train (s) | Inference (s) |
> | :--------- | :-------- | :------------ |
> | PartSLIP++ | —         | 5.83          |
> | Ours       | 2.25      | 1.46          |
>
> **Q3： How does the method perform on objects with significantly more complex part structures than those in PartNet-E?**
> **ANS:**  Our method is not designed for scene-level part segmentation (ScanNet) and image-based part segmentation (UDA-Part).  Extending the method to ScanNet requires both scene-level part annotations and non-trivial designs to handle thousands of frames all at once, which is out of the scope of this work and needs substantial efforts.  Below, we test on 3D part segmentation on ABC primitives and more complex shapes.
> 1\.  We selected 8 shapes for training and 100 shapes for evaluation from the ABC-Primitive dataset. We compared our method with PartSLIP++. However, as PartSLIP++ has not released its training code, we evaluated it in a zero-shot setting instead. Since the ABC dataset does not contain color information, we followed the approach used in PointCLIP-v2 by rendering the point clouds into depth images for processing. However, depth images are not a native input modality for models like GLIP, DINOv2, and SAM, leading to varying degrees of performance degradation for both PartSLIP++ and our method.
>
> | Method                 | mIoU  |
> | :--------------------- | :---- |
> | PartSLIP++ (zero-shot) | 3.68  |
> | Ours (few-shot)        | 40.13 |
>
> 2\. We note that the part annotations in Objaverse are quite inconsistent, and evaluate on the 3DComPat++\[1\] dataset for complex shapes with fine-grained part segmentation annotations. Due to limitations in time and computational resources, we selected three categories—**airplane**, **bicycle**, and **chair**—for evaluation. The number of fine-grained part labels for each of these categories is as follows: the airplane category contains 14 parts, the bicycle category contains 26, and the chair category contains 22, reflecting the complexity of these shapes\.
> We use the mesh data provided by 3DComPat++ and sample 100,000 points from each object as the experimental data. We compare our method with the representative baseline PartSLIP++. Since the training code of PartSLIP++ is not fully open-sourced, we evaluate it under the zero-shot setting. In contrast, our method is evaluated under both 1-shot and 8-shot settings. The evaluation metric reported in the table below is the mean Intersection-over-Union (mIoU).
>
> | Method                 | Airplane | Bicycle | Chair |
> | :--------------------- | :------- | :------ | :---- |
> | PartSLIP++ (zero-shot) | 9.48     | 2.60    | 4.87  |
> | Ours (1-shot)          | 45.34    | 26.26   | 27.21 |
> | Ours (8-shot)          | 65.38    | 54.17   | 42.61 |
>
> As shown in the table, our method performs well even when faced with more complex part structures, demonstrating strong generalization capability.
>
> **Q4:  How sensitive is the method to the number of views used for rendering?**
> **ANS:**  Both PartSLIP and 3-by-2 show that when the number of views exceeds 10, the performance gains begin to plateau. Based on this observation, we also adopt 10 views as the default configuration in our experiments.
> Our results are shown in the table below. When the number of views exceeds 10, the improvement in model performance significantly diminishes as the number of views increases. Conversely, when the number of views is fewer than 10, reducing the number of views leads to a substantial decline in model performance, highlighting the importance of sufficient multi-view information for the model’s effectiveness.
>
> | Category   |  15   |  12   |  10   |   9   |   8   |   7   |
> | :--------- | :---: | :---: | :---: | :---: | :---: | :---: |
> | KitchenPot | 85.25 | 85.61 | 85.20 | 80.15 | 79.85 | 73.49 |
> | Bottle     | 92.51 | 90.85 | 90.00 | 88.70 | 81.07 | 80.64 |
> | Chair      | 89.97 | 89.83 | 89.50 | 84.07 | 82.35 | 81.36 |
>
>
> \[1\] 3DCoMPaT^{++}: An improved Large-scale 3D Vision Dataset for Compositional Recognition
>
> **If you have any further question, please let us know. Thank you very much!**

---

> > ### Comment · Reviewer_URb2 · 2025-08-07
> >
> > Thank you to the author for the comprehensive clarifications and additional experiments. They effectively address most of my concerns.
> > The detailed runtime analysis provides valuable insights into your method's efficiency, and the experiments on ABC-Primitive and 3DComPat++ demonstrate promising generalization capabilities.
> >
> > Regarding the computational cost analysis, you mentioned that the primary preprocessing bottleneck comes from SAM segmentation of 10 multi-view images, taking 52.40 seconds. You suggest that "faster models could be adopted in the future to accelerate this process."
> > Have you experimented with any lightweight segmentation alternatives (e.g., FastSAM, MobileSAM) to SAM? If so, how does the trade-off between segmentation quality and preprocessing time impact the final part segmentation performance? This analysis would be valuable for practitioners considering deployment in time-sensitive applications.
> >
> > I am happy to increase my score if the authors can address the remaining point. Thank you again for thoroughly addressing my concerns.

---

> > > ### Author Response · Authors · 2025-08-08
> > >
> > > Thank you for your suggestion. We conducted additional experiments using **FastSAM** and **MobileSAM**. Due to time constraints, we selected three categories for testing. The table below summarizes the preprocessing time required to segment ten multi-view images and the part segmentation results.
> > >
> > > Among the tested methods, **FastSAM** demonstrated the highest efficiency in the preprocessing stage, owing to its dedicated optimization for full-instance segmentation. In contrast, **MobileSAM**, when used in *everything mode* for full-instance segmentation, exhibits only limited speedup over the original SAM. As explained in MobileSAM’s GitHub issue 24, although the encoder in MobileSAM is significantly faster, the overall runtime remains dominated by the decoder, which is executed 64×64 times for a grid of 64\*64 point prompts.
> > >
> > > | SAM | FastSAM | MobileSAM |
> > > | :---- | :---- | :---- |
> > > | 52.4S | 1.3S | 41.1S |
> > >
> > > In terms of segmentation accuracy, adopting FastSAM or MobileSAM results in minor performance degradation on 2/3 categories compared to using the original SAM.
> > >
> > > |  | SAM | FastSAM | MobileSAM |
> > > | :---- | :---- | :---- | :---- |
> > > | KitchenPot | 85.2 | 84.3 | 82.4 |
> > > | Bottle | 90.0 | 90.9 | 90.7 |
> > > | Chair | 89.5 | 87.7 | 87.8 |
> > >
> > > We are currently evaluating the full results of FastSAM and will update them accordingly.

---

> > > > ### Author Response · Authors · 2025-08-09
> > > >
> > > > We have completed the evaluation of FastSAM on all categories of the PartNetE dataset. Compared to the original SAM, using FastSAM results in approximately a 2% performance drop.
> > > >
> > > > | SAM | FastSAM |
> > > > | :---- | :---- |
> > > > | 72.8 | 70.9 |

---

> ### Comment · Area_Chair_i1QL · 2025-08-06
> **Reminder**
>
> Dear Reviewer,
>
> This is a friendly reminder to check the authors' rebuttal and adjust your rating if necessary. Thanks for your contributions to the NeurIPS reviewing process.
>
> Thanks,
>
> Your AC

---

### Note · Authors · 2025-08-13

# Final Remark

We would like to express our gratitude to all reviewers and ACs for their constructive suggestions and engaged discussions on the paper.

We are encouraged by the recognition of the following merits by reviewers.

* Reviewers URb2 and epFc commented on the sound motivation of using SAM segments as the basis for 3D understanding and the effectiveness of the graph-based propagation.
* Reviewers URb2 and v5oZ recognized the view-quality-aware unpooling module as a good solution to the issue of varying segment quality.
* Reviewers v5oZ, hPPy, and epFc agreed that our extensive experiments demonstrate the robustness of the method.
* Three out of four reviewers also appreciated the clear writing throughout the paper.

Our responses to reviewers’ concerns are well received by reviewers. Furthermore, Experiments with GLIP rigorously validated the performance improvements brought by the graph structure. Runtime analysis shows the SAM segment-based graph is highly efficient in representing a shape, leading to faster inference time.

All reviewers agreed that the rebuttal addressed most of their concerns. As a result, three reviewers gave positive ratings, and reviewer URb2 also considered raising their score. We will incorporate suggested results and writing improvements, as pointed by reviewers.

In a summary, SegGraph presents a novel SAM segment-based graph learning method for  few-shot 3D part segmentation with well-validated contributions and the SOTA results.

---

### Decision · Program_Chairs · 2025-09-17

**Decision:**

Accept (poster)

**Comment:**

This paper studies the few-shot 3D part segmentation problem. In the paper, the authors propose a SAM segment graph-based propagation method to explicitly learn geometric features encoded within SAM's segmentation masks to achieve few-shot 3D part segmentation. It was reviewed by four expert reviewers, and all of them recommended accepting this paper after the rebuttal period. Therefore, it is a clear acceptance.